# Integrated Bioinformatics-Based Subtractive Genomics Approach to Decipher the Therapeutic Drug Target and Its Possible Intervention against Brucellosis

**DOI:** 10.3390/bioengineering9110633

**Published:** 2022-11-01

**Authors:** Kanwal Khan, Munirah Sulaiman Othman Alhar, Muhammad Naseer Abbas, Syed Qamar Abbas, Mohsin Kazi, Saeed Ahmad Khan, Abdul Sadiq, Syed Shams ul Hassan, Simona Bungau, Khurshid Jalal

**Affiliations:** 1Dr. Panjwani Center for Molecular Medicine and Drug Research, International Center for Chemical and Biological Sciences, University of Karachi, Karachi City 75270, Pakistan; 2Department of Chemistry, College of Science, University of Ha’il, Ha’il 81451, Saudi Arabia; 3Department of Pharmacy, Kohat University of Science and Technology, Kohat 26000, Pakistan; 4Department of Pharmacy, Sarhad University of Science and Technology, Peshawar 25000, Pakistan; 5Department of Pharmaceutics, College of Pharmacy, P.O. Box-2457, King Saud University, Riyadh 11451, Saudi Arabia; 6Division of Molecular Pharmaceutics and Drug Delivery, The University of Texas at Austin, 2409 University Ave., Austin, TX 78712, USA; 7Department of Pharmacy, Faculty of Biological Sciences, University of Malakand, Chakdara 18000, Pakistan; 8Shanghai Key Laboratory for Molecular Engineering of Chiral Drugs, School of Pharmacy, Shanghai Jiao Tong University, Shanghai 200240, China; 9Department of Natural Product Chemistry, School of Pharmacy, Shanghai Jiao Tong University, Shanghai 200240, China; 10Department of Pharmacy, Faculty of Medicine and Pharmacy, University of Oradea, 410028 Oradea, Romania; 11HEJ Research Institute of Chemistry International Center for Chemical and Biological Sciences, University of Karachi, Karachi City 75270, Pakistan

**Keywords:** isocitrate lyase, *Brucella suis*, molecular docking, brucellosis

## Abstract

*Brucella suis*, one of the causative agents of brucellosis, is Gram-negative intracellular bacteria that may be found all over the globe and it is a significant facultative zoonotic pathogen found in livestock. It may adapt to a phagocytic environment, reproduce, and develop resistance to harmful environments inside host cells, which is a crucial part of the Brucella life cycle making it a worldwide menace. The molecular underpinnings of Brucella pathogenicity have been substantially elucidated due to comprehensive methods such as proteomics. Therefore, we aim to explore the complete *Brucella suis* proteome to prioritize the novel proteins as drug targets via subtractive proteo-genomics analysis, an effort to conjecture the existence of distinct pathways in the development of brucellosis. Consequently, 38 unique metabolic pathways having 503 proteins were observed while among these 503 proteins, the non-homologs (n = 421), essential (n = 350), drug-like (n = 114), virulence (n = 45), resistance (n = 42), and unique to pathogen proteins were retrieved from *Brucella suis*. The applied subsequent hierarchical shortlisting resulted in a protein, i.e., isocitrate lyase, that may act as potential drug target, which was finalized after the extensive literature survey. The interacting partners for these shortlisted drug targets were identified through the STRING database. Moreover, structure-based studies were also performed on isocitrate lyase to further analyze its function. For that purpose, ~18,000 ZINC compounds were screened to identify new potent drug candidates against isocitrate lyase for brucellosis. It resulted in the shortlisting of six compounds, i.e., ZINC95543764, ZINC02688148, ZINC20115475, ZINC04232055, ZINC04231816, and ZINC04259566 that potentially inhibit isocitrate lyase. However, the ADMET profiling showed that all compounds fulfill ADMET properties except for ZINC20115475 showing positive Ames activity; whereas, ZINC02688148, ZINC04259566, ZINC04232055, and ZINC04231816 showed hepatoxicity while all compounds were observed to have no skin sensitization. In light of these parameters, we recommend ZINC95543764 compound for further experimental studies. According to the present research, which uses subtractive genomics, proteins that might serve as therapeutic targets and potential lead options for eradicating brucellosis have been narrowed down.

## 1. Introduction

Brucellosis is an endemic disease, also known as Malta fever, Mediterranean fever, undulant fever, and Bang’s disease caused by bacterium, i.e., *Brucella* genus (almost 12 species) belonging to a family of *Brucellaceae*, class *Alphaproteobacteria,* infecting both animals and humans [1]. Species of this genus are Gram-negative intracellular facultative pathogens. Brucellosis is characterized as an acute fever illness [2], associated with various symptoms in humans such as liver and spleen disorders, reproductive abnormalities, neurological problems, and heart-related problems, and also have been classified as a potential bioterrorism agent [3,4,5]. Brucellosis remains endemic in various emerging countries in Asia, Africa, Middle East, and South America, where screening of livestock and vaccination fails to control and exterminate the disease [6].

The world’s largest animal disease control agencies such as the Food and Agricultural Organization of the United Nations (FAO) and the World Organization for Animal Health (WOAH) all consider brucellosis to be highly contagious and warn about the disease management through the Global Early Warning System for Major Animal Diseases (GLEWS). However, human brucellosis is still the most prevalent animal-to-human transmission disease in the world [7,8]. The socio-economic effect of brucellosis is huge and larger in developing nations than in industrialized ones, with an estimated 3.5 billion people at risk of infection with one or more *Brucella* spp. and a high morbidity rate in humans and animals [9]. According to Hull [10], there are 500,000 cases of human brucellosis reported per year around the globe due to their ability to survive and multiply within the host phagocytotic and non-phagocytotic cells. Surprisingly, *Brucella* did not show classical virulence mechanisms such as producing cytolysins, plasmids, fimbria, an exotoxin, exoenzymes, and drug-resistant forms. However, *Brucella* bacteria comprise various virulence factors such as lipopolysaccharide (LPS) [11], β-cyclic glucan [12], outer membrane proteins (Omps) [13,14], MucR [15], T4SS secretion system, and BvrR/BvrS system which have been identified, which permit the *Brucella* to interact with the host cell [16]. Also crucial to *Brucella* pathogenesis is a complex of VirB proteins and their five effectors that use the T4SS to control the host cell’s inflammatory response and vesicle trafficking [17,18]. Due to our current lack of understanding of *Brucella suis* genomes, it is challenging to create a novel species-specific therapeutic molecule experimentally [19]. Protective immunological interventions and the development of novel treatment techniques are of great importance in the fight against antibiotic resistance.

However, screening hundreds of macromolecules and the in vivo studies that follow are time-consuming and resource-intensive tasks in the drug development process. To compare the selectivity and specificity of potential therapeutic targets, subtractive genomics is one of the most used computational methods. Many researchers [20,21,22,23] have extensively reported the application of subtractive genomics techniques against distinct pathogenic strains for the discovery and identification of new species-specific therapeutic targets.

Our present work employed subtractive genomic analysis against a comprehensive examination of the *Brucella suis* strain’s whole proteome in order to identify potential therapeutic targets [24,25]. Following this research, we may conclude that *Brucella suis*’ resulting proteins may be the most promising therapeutic targets and shortlisted candidates that can be used for establishing a universal drug molecule that may provide a basic pipeline for drug discovery against brucellosis. Virtual screening against a chosen target was also performed utilizing the Zinc library to discover promising inhibitors for the treatment of brucellosis. The ZINC database is a collection of commercially accessible chemicals with ready-to-screen biologically relevant descriptions. Each molecule in the library is prepared for docking using a variety of common docking programs and includes details on the vendor from whom it may be purchased.

## 2. Results

The primary objective of this research was to use a subtractive genomic method for the whole proteome of *Brucella suis* 1330 in order to discover potential new therapeutic targets against this pathogen. Table 1 provides a brief overview of the study’s methodology and results.

### 2.1. Unique Metabolic Pathways Analysis

Complete metabolic pathways of both *B. suis* (109 pathways) and human host (330 pathways) were downloaded from the Kyoto Encyclopedia of Gene and Genome (KEGG) server. We compared both the human and *B. suis* metabolic pathways manually to find both the common and unique metabolic pathways. The results showed that 71 pathways were common (Appendix A) both in humans and *Brucella suis* and 38 pathways were unique to *B. suis*. We selected unique metabolic pathways which were comprised of proteins unique to *B. suis* and excluded the common metabolic pathways. Consequently, the total unique metabolic pathways consisted of 503 proteins (Figure 1).

### 2.2. Prioritization of Non-Homologous Proteins

Homologous proteins have developed over time in shared bacterial and human cellular systems. As a result, it is important for therapeutics designed to bind pathogen target proteins to not cause unwanted reactions in the host by binding their homologous proteins. We subjected 503 unique metabolic pathways of *B. suis* to run a BLASTp with a cutoff value 10^−3^ against the whole proteome of humans to identify only non-homologous proteins for novel drug target prioritization. We selected only non-homologous proteins to avoid the undesirable side effects of the drug. The BLASTp results showed 82 proteins were homologous to the human host with high similarity with the human proteome and were excluded. The remaining 421 non-homologous proteins were analyzed in the next step.

### 2.3. Identification of the Essential Proteins

The Database of Essential Genes (DEG) provides complete information of the essentiality of proteins of the bacteria determined from the experimental analysis. Those proteins having high similarity with proteins of DEG were selected as essential proteins. We ran a BLASTp of non-homologous 421 proteins against the DEG database with a default parameter 60% sequence identity. The results showed that 350 proteins were essential for the survival of *B. suis*. Targeting the essential proteins with a recommended drug may halt the vital activity of *B. suis*.

### 2.4. Druggability Potential of Essential Proteins

The current study was further improved by looking for the druggability potential of the shortlisted protein sequences of *B. suis*. We ran a BLASTp of the non-homologous essential proteins against the DrugBank database with an E-value 10^−5^. This analysis led to the identification of 114 proteins that may act as drug targets approved by the Federal Drug Authority (FDA) in a DrugBank database with its recommended IDs. As the research concentrated on the discovery of powerful drug targets, these 114 proteins exhibited substantial similarities with the authorized FDA drug targets (experimentally demonstrated data for drug targets) and were chosen for future investigation. However, the excluded 136 may potentially function as therapeutic targets since they are important, non-homologous proteins.

### 2.5. Predicting the Virulence Factor (Proteins) of B. Suis

The VFDB database comes up with complete information of protein virulence. The VFDB explored the importance of virulent proteins in disease progression. The VFDB results revealed that 45 proteins out of 136 were correlated with the virulence of *B. suis*. However, these 45 proteins can be used as a novel and potent drug target against *B. suis* strain 1330.

### 2.6. Identification of the Resistance Proteins

The pathogens resistant to drugs are more hassle to treat the disease, which requires a higher dose that shows a diverse effect in patients. The pathogen acquired with drug resistance is due to continuous exposure to a drug or the drug used in a higher dose. We subjected the FASTA sequence of shortlisted virulent proteins to the ARG-ANNOT V6 online tool. The results showed that 42 out of 45 proteins were correlated with the resistivity of a pathogen. These 42 proteins are mostly involved in the degradation and efflux of numerous drugs. However, these 45 proteins can be used as a potential drug target. The list of these 42 drug target-like proteins is provided in Appendix A.

### 2.7. Prediction of Subcellular Localization

All proteins require a specific location for their optimal function. Transporting proteins to an unspecified region may result in server diseases [26]. All the organism’s cells have a special compartment. After synthesis and modification, all the proteins must be destined to the appropriate compartment to do their proper function. Subcellular localization affects the protein function by monitoring access to and availability of molecular interaction partners [27]. All FASTA sequences of shortlisted proteins were subjected to the online server PSORTb for the prediction of subcellular localizations of proteins. The subcellular localization of proteins is determined on the basis of overall amino acid composition of proteins, sequence homology and motifs, known targeting sequence, and combined information from the above methods [28]. Based on the subcellular localization of proteins, we design vaccine and drugs against the specific localized protein target. The cytoplasmic proteins may act as a drug target and outer membrane proteins may act as a vaccine target. The results of current study showed ~23 cytoplasmic, ~14 periplasmic, ~6 outer membrane, and ~4 inner membrane proteins. Figure 2A represents the graphical sketch of subcellular localization.

### 2.8. Significant and Novel Drug Targets Prediction

It has been reported that cytoplasmic proteins are an excellent therapeutic target and may be readily targeted with drugs [29]. Additionally, ~70 percent of FDA-approved medications are claimed to target enzymatic proteins because of their substantial participation in numerous pathways. Finally, among 42 candidate therapeutic targets, one protein was identified as an essential, non-homologous, druggable target against *Brucella suis* i.e., isocitrate lyase. Based on its cytoplasmic subcellular distribution, length > 100 amino acids, enzymatic nature, and role in important metabolic pathways, this discovered protein was advanced to additional structure-based investigations.

### 2.9. Isocitrate lyase

The enzyme isocitrate lyase (EC 4.1.3.1), often known as ICL, is a key enzyme in the glyoxylate cycle. It plays a role in metabolic adaptability to environmental changes. To replenish the tricarboxylic acid cycle during development on fatty acid substrates, this enzyme catalysis is the reversible synthesis of succinate and glyoxylate from isocitrate, a critical step in the glyoxylate cycle. The protein helps in the catalysis of reactions such as,
 ICL
D-*threo*-isocitrate ⟶ glyoxylate + succinate

It is widely studied as a potential drug target against MTB for the designing of novel therapeutics [30,31], *Candida albicans* [32], *Paracoccidioides brasiliensis* [33] and for the development of new anti-buruli ulcer natural products [34]. However, ICL protein has never been explored as a drug target for *B. suis,* and thus, in the current study it is proposed as a potential novel drug target.

### 2.10. Protein–Protein Interaction Analysis

The protein–protein interaction (PPI) and their functional annotation make the pillar of cellular machinery as it controls a wide range of biological processes [35]. Revealing PPI information helps us in the identification of drug targets [36,37]. One has to identify various interactions and govern the outcome of the interactions to precisely understand the PPI and their importance in the cell [38]. In the current study, we used the STRING database for the prediction of PPI of shortlisted protein, i.e., isocitrate lyase.

Similarly, an empty and filled node represents protein with uncharacterized and characterized 3D or predicted structure, respectively. The STRING results showed that our shortlisted protein may act as hub protein because they make various interactions with other nearby proteins to perform a crucial function. If we target isocitrate lyase, it may halt the function of other interactor proteins since proteins work in a group [39].

We uploaded the isocitrate lyase NCBI ID (WP_004691511.1) to the STRING database online server and found the interactions with other proteins in the *B. suis* biovar 1330. The isocitrate lyase was represented by BMEI0409 and it mediates interactions with other proteins nearby like: malate synthase G, involved in the glycolate utilization (glcB); aconitate hydratase (BMEI1855), (BMEII0009), (BMEI1952), (BMEII1061), (BMEI1939), (BMEII1060), (BMEII1062), (BMEII1064); and ureidoglycolate lyase (allA) and (BMEI0799). The PPI results showed that isocitrate lyase had a total number of nodes 11, average node numbers 7.09, average local clustering coefficient 0.95, the total number of edges 39, PPI enrichment *p*-value 3.81 × 10^−11^, and 11 expected number of edges as shown in (Figure 2B). These proteins are involved in a variety of critical functions. By inhibiting the isocitrate lyase, the remaining interactor proteins may also lose their function. Therefore, isocitrate lyase can be proposed safely as a potential drug target.

### 2.11. Comparative Structure Homology Modeling and Validation

The 3D structure of isocitrate lyase was constructed through an online bioinformatics tool called Phyre2. This server uses unconventional remote homology prediction approaches to build 3D models of desired proteins, predict active sites, and analyze the effect of amino-acid variants (e.g., nsSNPs) of a protein sequence. We retrieved the FASTA sequence of isocitrate lyase with an NCBI accession number (WP_004691511.1) and uploaded it to the Phyre2 server. The Phyre2 results give us different PDB structures of proteins based on sequence similarities and percent identity. The Phyre2 used a different template and showed different percent identity and confidence scores. The templates 6EDW, 3E5B, and 1F61 all had confidence scores of 100 and percent identity 42, 99, and 66, respectively. We selected the modeled structure which used the 3E5B as a template (Figure 3A).

The successfully constructed structure of isocitrate lyase was confirmed with online bioinformatics tool, Procheck. It checks the stereochemical properties of modeled protein and residue-by-residue geometry and compares them with ‘ideal’ values obtained from a database of well-refined and well-defined high-resolution protein 3D structures in the Protein Data Bank. The Procheck results showed that 90.1%, 8.5%, 1.1%, and 0.3% of residues lie in most favored regions, additionally allowed regions, generously allowed regions, and disallowed regions, respectively, as shown in Figure 3B. Likewise, Phyre2 also predicts the secondary structure of proteins based on the primary sequence of amino acid. The Phyre2 results disclosed that the secondary structure of isocitrate lyase is comprised of 53% of alpha-helix, 22% disordered regions, and 8% of beta-strand (Figure 3C).

Nevertheless, in PDB databases, three isocitrate lyase structures were available from *Brucella abortus* 2308 (PDB ID: 7RBX), *Brucella melitensis* (PDB ID: 3E5B), and *Brucella melitensis* (P43212) (PDB ID: 3EOL). The multiple sequence alignment was performed for these PDB sequences with modeled protein to study the sequence’s similarities among them. It showed that WP_004691511.1 protein is 93.3% similar to 7RBX and 3E5B, while 99.07% to 3EOL, resulting in almost 99% similarities. The sequences vary at 153 (G to D), 156 (N to D), 330 (C to R), and 400 (M to V amino acids) positions (Figure 4A). The ICL protein structures of all three strains were superimposed individually on the template strand of *B. suis.* When aligned to the template, the ICL proteins from *B. suis* strains showed an overall RMSD value of 1.99 Å (Figure 4B). This showed that the available protein structure of ICL from these strains can also be used for the *B. suis* ICL.

### 2.12. Active Site and Ligand Prediction of Modeled Protein

The ligand binding to its active site is required for a protein to function properly. A huge number of bioinformatics tools are available to predict the ligand binding site and it is the first step in understanding the function of the protein and facilitating the docking and virtual screening-based drug discoveries. Experimentally or computationally modeled 3D structure of a protein is a prerequisite for the prediction of the ligand binding site. We uploaded the 3D modeled structure of isocitrate lyase to the Molecular Operating Environment (MOE) standalone tool. The MOE identified a ligand binding site on the template that makes use of a similar 3D structure with a known ligand binding site, based on geometry computation to identify binding pockets. The result gives us different active sites with functional residues. Based on high energy, we selected the first ligand binding site (Figure 5A) having active site residues.

The prediction of ligands of proteins is the most challenging job in the field of biochemistry, pharmacy, and protein chemistry. The biochemical function of a protein can be elucidated by comparing the protein with a well-refined 3D structure and known functional ligand, although a similar folding pattern of protein is not a guarantee for a similar function. Likewise, the same biochemical characteristics may be shown by proteins having different folding patterns. Now it seems clear that the binding site of the protein is the primary determinant of protein function rather than its folding pattern [40,41]. In the current study, we used the ProBis online tool for ligands prediction. A 3D structure of isocitrate lyase was uploaded to the ProBis tool. The results showed that isocitrate lyase had high sequence and structure homology with isocitrate lyase from *Brucella melitensis* complex with isocitric acid (ICT) as ligand with a PDB ID (3P0X) (Figure 5B).

### 2.13. Molecular Docking Study Virtual Screening

Plenty of bioinformatics tools have been developed and widely used for the molecular docking of the ligand with protein, particularly for the drug discovery approach [39,42]. The MOE, which helps to characterize, visualize, and evaluate protein interactions with ligands or other proteins, was utilized in the current study. The MOE is a very user-friendly tool for molecular docking study among all docking tools. It shows a good graphical representation of ligand active site interaction. In MOE, receptor-ligand binding affinities with all possible binding geometries are selected based on a numerical value called an S-score. A low S-score shows the highly docked compound. Molecular docking of isocitrate lyase was performed with a standalone bioinformatics tool (i.e., MOE). The active site was found as discussed earlier. The protein was prepared for docking study followed by protonating its 3D structure, along with energy minimization while setting all the parameters at default values followed by ligand docking with isocitrate lyase using MOE. The result showed the binding of isocitric acid ligand (ICT) with the protein with five different conformations and orientations. We selected conformation 1 of the ligand on the basis of high binding energy, i.e., −6.9527, whereas the lowest binding score was −4.57. The docking result showed that Glu 281, Ser 283, and Ala 227 mediate hydrogen bonds, whereas Lys 191 form three ionic interactions with the ligand as shown in Figure 6.

The 18,000 ZINC library was subjected to virtual screening using rigorous docking to the active site of isocitrate lyase. It gave several docked conformations of compounds that were defined by docking scores. Lower binding affinity compounds (i.e., those with a binding affinity less than the reference ICT inhibitor at −6.95 kcal/mol) were excluded from assessment as potential hit candidates. More than 10,000 compounds were found to have greater binding energies compared to ICT, which ranged in the limit of between −7.0–12.0 kcal/mol for the screening results (Figure 7A (brown hue)). These 10,000 molecules suggest that ICL inhibition may be a potential lead. Due to their lower binding affinity than ICT inhibitors, the compounds were chosen for further exploration in this work because of their significant inhibitory effect against ICL (Figure 7B). There were only six possible therapeutic candidates against ICL, which were ZINC95543764, ZINC02688148, ZINC20115475, ZINC04232055, ZINC04231816, and ZINC04259566 to inhibit *B. suis* 1330 serovar.

### 2.14. Interaction Analysis of Shortlisted Compounds with ICL

Shortlisted compounds were examined using post-molecular docking interaction analysis to better understand ICL’s pharmacological activity and binding mechanism. Each ligand had several interactions with the receptor in molecular docking research. ZINC20115475 > ZINC02688148 > ZINC04231816 > ZINC04232055 > ZINC04259566 > ZINC95543764 is the docking rank order based on docking score.

ZINC20115475 was shown to have a binding energy of −10.9 kcal/mol in a docking study. The six-ring aromatic ring mediates two pi and hydrogen bonds with Gln72, and Gly353 each with a bond distance of 3.63 and 3.59, along with an energy of −0.7, correspondingly (Figure 8A).

ZINC02688148’s binding score was found to be −9.8 kcal/mol. Through its aromatic ring, which has a bond distance of 3.78 and energy of −0.7 kcal/mol, it binds Lys317 to one pi-hydrogen (Figure 8B).

With a binding score of −9.0 kcal/mol, ZINC04231816 was well-suited for ICL’s binding pocket. A single aromatic-hydrogen interaction between Ala386 and the aromatic ring mediates four hydrogen bonds as hydrogen acceptors from Arg116, Arg123, and Gln72 through its O13, O14, with a bond distance 2.91–3.14 and an energy range of −1.3 to −4.0 kcal/mol (Figure 8C).

The ZINC04232055 was found to mediate four hydrogen bonds as hydrogen acceptors with Gln72, Ala386, Arg123, and Arg119. It resulted in binding energy of −8.6 kcal/mol (Figure 9A).

ZINC04259566, on the other hand, was observed to initiate two hydrogen bonds from Arg116 and Thr120 as hydrogen acceptors. One pi-cation interaction between the aromatic ring and Arg116 was also found. The binding score was −8.5 kcal/mol (Figure 9B).

ZINC95543764 docked with ICL at −8.1 kcal/mol, and its aromatic ring mediates a single pi-hydrogen bond with the Leu372 residue (Figure 9C). A description of binding interactions formed inside ICL’s active cavity by the selected compounds is shown in Table 2.

Additionally, docking studies predicted the Ki (inhibitory concentration) of the shortlisted compounds. The probability that a drug may inhibit an enzyme and result in a clinically meaningful pharmacological interaction with an enzyme substrate is estimated using the Ki. The Ki has often been computed in correlation to the body’s inhibitor concentration, and it serves as the basis for a program or particular drug information websites to assess whether a drug is an inhibitor or not. The predicted *K*_i_ values for the lead compounds were in the range of 9.87–1020 nm (Table 3). The binding energy and p*K*_i_ of the compounds showed a relatively strong correlation among them (*r*^2^ = 0.99) (Figure 10).

### 2.15. ADMET Profiling

A critical step in the identification of new drugs is the characterization of ADMET characteristics, since it saves time and money during clinical trials [43,44]. Based on drug-likeness, ADME profiles, and blood–brain barriers (BBB) analyses, the online SwissADME was utilized to compute the pharmacokinetic parameters of the nominated compounds. The Lipinski Rule of 5 was utilized as a basis for the drug-likeness characterization. Drug-like compounds should have molecular weights below 500 amu, hydrogen bond acceptor sites below 10, hydrogen bond donors below 5, and lipophilicity values (LogP) below 5. According to RO5, this is the recommended molecular weight range for orally delivered drugs and compounds. Consequently, it was observed that ZINC20115475, ZINC02688148, and ZINC04259566 fulfilled the Rule of Five (RO5) while ZINC95543764, ZINC04232055, ZINC04231816 showed one violation. However, these six compounds were observed to have no BBB permeability. Drug-likeness standards were followed by compounds that advanced to the next step.

In order to estimate the preliminary ADME properties, the pkcsm tool was used. These properties were characterized by the solubility in pure water (mg L), absorption in the gastrointestinal tract (HIA), permeability, inhibition of the liver enzyme CYP 2C19, inhibition of the CYP 2C9 enzyme, inhibition of the CYP 2D6 enzyme, inhibition of the CYP 3A4 enzyme, and inhibition of the Caco2 permeability. These compounds showed water solubility within the range of −2 to −5 with the ZINC02688148 compound having high water solubility. Caco2 permeability was observed as −0.025 to 0.94 with ZINC04259566 having high cell permeability and ZINC95543764 having the least permeability. Moreover, all compounds showed good to potent HIA permeability (Table 4). Based on these activities, we recommend ZINC95543764 compound for further experimental studies.

Additionally, Ames mutagenesis study used the pkcsm tool to identify the chemicals and predict their toxicity, max. tolerated dose (human), minnow toxicity, *T. Pyriformis* toxicity, oral rat acute toxicity (LD_50_), hepatotoxic, and skin sensitization evaluation. Consequently, all compounds showed a negative Ames test except ZINC20115475. This means that these molecules do not cause mutagenicity, as we know that the basis for the prediction of toxicity from a chemical structure is that the properties of a chemical are implicit in its molecular structure. Biological activity can be expressed as a function of partition and reactivity, that is, for a chemical to be able to express its toxicity, it must be transported from its site of administration to its site of action, and then it must bind to or react with its receptor or target. This process may also involve the metabolic transformation of the chemical. So according to that, the compound ZINC20115475 may, because of its chemical structure, cause mutagenicity and is indicated as positive for Ames test. Therefore, we excluded this compound from our further studies due to violation of Ames toxicity. On the other hand, ZINC02688148, ZINC04259566, ZINC04232055, and ZINC04231816 showed hepatoxicity while all compounds were observed to have no skin sensitization. *T. pyriformis* showed maximum tolerance to ZINC20115475 (0.367 log ug/L), while less tolerance was seen for the remaining compounds. The detailed information is shown in Table 5.

### 2.16. Conservancy Analysis of Predicted Sequences with Other Strains

The extent of the pharmacological spectrum over the whole homologous bacterial population might be inferred through comparison of the predicted sequences’ conservation pattern with other strains that are utilized conventionally. Therefore, the conservancy analysis was performed for ICL protein (WP_004691511.1) to examine the ICL presence in other strains. The BLASTp of ICL resulted in the local alignment of protein with six different *Brucella* strains ICL proteins with 99% identity, i.e., WP_004691058.1 (99.77%), WP_006071281.1 (99.53%), WP_006201131.1 (99.30%), WP_004689882.1 (99.53%), WP_006279211.1 (99.30%), and WP_004684105.1 (99.07%) showed that these proteins are 100–98% conservation among them (Table 6). The predicted protein always should be conserved in a wide range of bacterial strains for ensuring an effective drug and vaccine target against a specific bacterial pathogen [45]. Here, ‘isocitrate lyase ICL’ displayed a higher conservancy pattern with homologous proteins of other *Brucella suis* strains from different geographical locations. Hence, ‘ICL protein’ could be a potent broad-spectrum drug target for *Brucella suis*.

## 3. Materials and Methods

The present investigation used a subtractive genomic approach to identify pharmacological targets unique to *Brucella suis* [46]. Figure 11 demonstrates the whole protocol used in this investigation, whereas the research follow-up with analysis in detail is given below sequentially.

### 3.1. Data Retrieval

The human proteome from the UNIPROT database and the *B. suis* proteome from the National Centre for Biotechnology Information (NCBI) database [47] were obtained (Table 7. Proteins’ essentiality was determined using the DEG database, and druggable proteins were found using DrugBank. In contrast, both human and *Brucella* metabolic pathways were retrieved from the KEGG server.

### 3.2. Phase I: Subtractive Genomic Analysis

The subtractive genome analysis has been described as a promising, efficient, and novel in silico technique utilized to find therapeutic drug targets [48]. The subtractive genome analysis technique is based on sorting of the unique protein (missing in the human host) from the complete proteome of both host and pathogen in order to improve drug development pipeline by avoiding the cross-reactivity of the medications with human-host proteome. Steps included in the subtractive genome study are discovering non-paralogous sequence, non-homologous sequence, predicting essential proteins, finding proteins participate in metabolic pathways, and identification of proteins that may serve as a therapeutic target.

### 3.3. Prioritization of Pathogen-Specific Metabolic Pathways

The human-host and *Brucella suis* metabolic pathway databases KEGG [49] and Automatic Annotation Server (KAAS) were utilized for the study. We used the three-letter KEGG organism codes ‘bms’ for *B. suis* and ‘has’ for H. sapiens to extract the metabolic pathways ID and associated information from the KEGG database. There were just a few metabolic pathways that could be found in both; thus, they were clustered together as a single metabolic pathway. The NCBI database was used to get the FASTA sequence of proteins involved in the unique metabolic pathways of *B. suis*.

### 3.4. Non-Homologous Proteins Identification

A BLASTp search against the human proteome, with a cutoff value of 0.0001, was used to analyze the unique metabolic pathways proteins (E-value 10**^−^**^3^). Excluded proteins with high sequence similarity (>80%) to the human proteome were retrieved in the following phase of subtractive genomic analysis, and the remaining proteins were further studied.

### 3.5. Prioritization of Essential Proteins among the Whole Proteome of Brucella suis

In order to prioritize the essential proteins of the *B. suis,* the proteins were analyzed via BLASTp with a threshold value of 10**^−^**^5^, against the Database of Essential Gene (DEG) [50] consisting of proteins responsible for the survival of the organisms. Those proteins which had sequence similarity with essential proteins of the DEG were further analyzed and the non-essential proteins were excluded.

### 3.6. Druggability of Selected Sequences

All the essential, non-homologous proteins were then evaluated by BLASTp against the Drug Bank database [51]. The Drug Bank consists of a number of drug targets like proteins with specific target ID which are authorized by Food and Drug Administration (FDA). The default parameter of E-value 10**^−^**^5^ was applied to BLAST the proteins against the Drug Bank to find the novel drug targets.

### 3.7. Determination of Virulence Factor (Proteins)

The virulence of proteins helps bacteria to destroy the host-immune system with the help of colonization and invading the host immune cell and, as a result, the disease is caused. For the determination of the virulence of proteins, VFDB (virulence factor of pathogenic bacteria) online database [52] was used. The shortlisted proteins of *B. suis* were subjected to BLAST against the VFDB.

### 3.8. Identification of Resistance Proteins

The Antibiotic Resistance Gene-ANNOTation V6 (ARG-ANNOT V6) tool was used for the prediction of novel resistance protein sequences from the whole genome and proteome of a pathogen. All the resistant protein data were collected and analyzed from different experimental published work and different online sources, and protein sequences were retrieved from the NCBI database. The shortlisted proteins’ FASTA sequence were then subjected to BLAST against the resistance proteins of the ARG-ANNOT V6 database with a threshold of 10**^−^**^5^ [53].

### 3.9. Subcellular Localization and Protein–Protein Interaction Analysis of Shortlisted Proteins

All the shortlisted proteins were then subjected to PSORTb version 3.0.2 [54] and the Cello v.2.5 [55] online tool in order to identify subcellular localization. Subcellular localization (SCI BLAST) uses the PSORTb and Cello v.2.5 web servers to BLAST all of the nominated proteins. There is the cytoplasm, membrane, periplasmic membrane, and extracellular space in the subcellular localization.

The protein–protein interaction was evaluated via a bioinformatics online database called STRING [56]. It is comprised of experimentally known and predicted physical and (direct) and functional (indirect) interactions of proteins with each other in close proximity.

### 3.10. Structure Modeling and Validation

Since the three-dimensional (3D) PDB structure of isocitrate lyase out of three shortlisted proteins was not present in the Protein Databank (PDB) [57], the 3D structure of isocitrate lyase was constructed by online server, Phyre2 [58]. Only the FASTA sequence is needed for the construction of the 3D structure of protein which was retrieved from the NCBI database.

Moreover, different bioinformatics tools such as Procheck and PsiPred were used for the validation of constructed 3D structure of a protein. The Procheck server examines each residue conformation and geometry of the 3D protein structure [59]. It uses the PDB’s well-defined and well-refined high-resolution protein structure to determine the best possible values for key protein properties. There is an online server called Phyre2 that predicts protein secondary structure (random coils, sheets, and helices). A protein’s secondary structure was predicted by Phyre2 using PsiPred and its amino acid sequence.

### 3.11. Prediction of Active Site and Potential Ligand

For the identification of active site, MOE software was used. The prediction of active site residues relied on detecting the conserved regions beyond the family [60,61], powerful sequence-based scoring functions [62], information from the well define 3D structure, and analyzing the features such as the geometry of the amino acid residues, [63] electrostatic and chemical properties [64].

A computational online tool ProBis (protein binding site) was used for the ligand prediction. The Probis tool is comprised of a database for the foretelling of the ligand based on the sequence similarity with deposited 3D structure in PDB database [65].

### 3.12. Molecular Docking and Virtual Screening

For the docking experiment to proceed, the protein’s 3D structure must be available. The isocitrate lyase receptor was a three-dimensional model of the enzyme’s structure, and the inhibitor was a predicted ligand. The protein and ligand complex were thus selected and prepared for docking experiments. Ligand and other heteroatoms were removed from the protein during the filtration process (including water molecules). AutoDock v4.2 [66] was used for the remainder of the protein production. All hydrogen addition, the fusion of non-polar hydrogen atoms, and the addition of Kollman charges are components of the receptor preparation. A local shell was used to store the receptor when it was prepared. AutoDock was used for molecular docking and the standard docking approach was followed. For the ligand’s docking and implementation, 250 times Lamarckian GA settings yielded 27,000 maximum generation generations and 2,500,000 evaluations. The re-docking was performed to assess the performance of the docking program for its capability of reproducing the same crystal conformation of the bound ligand. The X, Y, and *Z*-axis grid points were set to 50, 52, and 50, correspondingly, with grid center values of 42.32, 64.849, and 13.3979.

The ZINC library of 18,000 molecules was recovered in SDF format and kept in local bash depositories to continue identifying potent drug-like molecules. A 3D PDB file was generated from the downloaded 2D compound file using Open Babel [67]. A FROG2 was used to minimize the ligand library’s energy using the MMFF94 force field and steepest descent technique for 1500 steps. Additional steps included adding gastieger charges to the ligand library, rotating all rotatable bonds in AutoDock, and then saving the optimized ligand library in PDBQT format for additional virtual screening. Vina split was used to divide the built PDBQT library into the corresponding files. The redocking experiment’s settings and grid box configuration were carried over to the virtual screening.

### 3.13. ADMET Profiling

Safe drug therapy assessment is a critical concern in the drug development process. Early detection of potent pharmaceutical toxicity and side effects is critical for minimizing drug development time and price [68]. Therefore, pharmacokinetic parameters such as absorption, distribution, metabolism, and excretion (ADME) for the shortlisted drug-like compounds were examined using the SwissADME tool [69]. The pkCSM tool (http://biosig.unimelb.edu.au/pkcsm/ accessed on 14 Februry 2022) was then used to investigate the compounds’ toxicity profile, minimal human side effects, immunotoxicity, mutagenicity, teratogenicity, neurotoxicity, increased penetration, and carcinogenicity.

### 3.14. Conservancy Analysis of Predicted Sequences with Other Strains

The extent of the pharmacological spectrum over the complete homologous bacterial population may be inferred by comparing the conservation pattern of the predicted sequences with other conventionally utilized strains. The BLASTp software on the NCBI website was used to conduct a conservancy analysis of the projected drug target sequences. Here, we performed a protein–protein BLAST with all default settings except for the organism option, where we specified the *B. suis* 1330 strain.

## 4. Discussion

Current work culminates into a comprehensive understanding of outlined objectives by systematic in silico application of drug design modules against *B. suis*. *Brucella* is a zoonotic, Gram-negative intracellular pathogen that infects cattle, small ruminants, dogs, and pigs, whereas non-zoonotic *Brucella* species may infect many animals, including humans [70]. Brucellosis in animals may cause miscarriage, reduced milk supply, reproductive issues, and hygromas [70,71]. To stop disease, infected animals are killed. *B. melitensis* and *B. abortus* have cattle vaccinations, while *B. suis* does not [71,72]. Due to its high infection rate and economic burden on agriculture and health sectors, effective therapies such as novel medications and vaccines are needed.

Computational biology facilitates the study and interpretation of large amounts of data [73]. Predictions that have never been made before are being made using data mining techniques. Several genome annotations have emerged in the post-genomic age that is beyond the comprehension of human beings [73]. Computational biology is used to find prospective therapeutic targets in the realm of infectious diseases. Reducing genetic data to its essential components is one such approach, i.e., subtractive genomics [74]. Genomic analysis at its core is a promising strategy for tackling infectious pathogens that affect several species. Pathogens rely on their conserved-core genes for their very life [74], therefore developing a treatment that specifically targets these genes might be an effective strategy for eliminating bacterial infection. Research on *Escherichia albertii* [75], *A. baumannii* [76], *Mycobacterium* spp. [77], etc., provided evidence for this claim. The study against these pathogens identifies the drug target from the core genome that are previously not highlighted.

Therefore, the current study applied the subtractive genomic approach for the prioritization of novel targets against *B. suis* (Figure 11). The study characterized 42 putative druggable proteins from whole proteome subtractive analysis of the *B. suis* 1330 strain. Using this method, we were able to identify an essential, non-homologous protein linked with the pathogen and postulate that it may be a therapeutic target, i.e., isocitrate lyase (ICL) was discovered to be present in the unique metabolic pathway (Table 1) having unique protein only in the *B. suis* genome, as shown in Table 2.

To avoid the two decarboxylation stages of the tricarboxylic acid (TCA) cycle, isocitrate lyases (ICLs; isoforms 1 and 2) catalyze the reversible conversion of isocitrate to glyoxylate and succinate in the glyoxylate cycle [78]. The previously unexplored genome of B. suis has marked the identification of ICL as one of the key findings of the study. ICL has been classified as a promiscuous target for *M. tuberculosis* due to its persistence and constitutes a target for antituberculotic agents [31]. Similarly, Chung et al. also target the ICL protein for *Candida albicans* [79]. ICL has recently been widely targeted for the cure of numerous bacterial infections and thus can also be used as a potential target for *B. suis* 1330. Nevertheless, the conservancy analysis validated it as a single target for all *Brucella* spp.

The binding site and binding ligands were also predicted to have more detailed knowledge regarding protein structure. The ICL structure was modeled and validated using Phyre2, Procheck, and PSIPRED tools. Lastly, PSORTb analysis showed the suitability of the protein as a drug target as it was cytoplasmic. Moreover, using the molecular docking approach for the identified drug target, we identified six compounds, i.e., ZINC95543764, ZINC02688148, ZINC20115475, ZINC04232055, ZINC04231816, and ZINC04259566 (from shortlisted 10,000 compounds having high binding affinity compared to control ICT) that potentially inhibits isocitrate lyase as the most favorable interactions with the target active site residues. The predicted Ki and binding affinity correlation showed that these compounds might be able to block or dampen a biological response by competitively binding and thus blocking ICL activity.

Furthermore, CDC reported that major *brucellosis* occur due to the resistance of these strains. Therefore, it is imperative to have a protective plan in case of major future outbreaks. The vast array of information regarding the proteomes of *Brucella suis* can be clinically implemented for possible pathogen identification and its therapy against ICL. The obtained results can be manipulated to accelerate drug designing and gain further knowledge of pharmacogenomics in the treatment of brucellosis. Moreover, the applied subtractive genomics can aid in the identification of proteins targeted by existing FDA approved targets other than *brucella*.

This study may facilitate future researchers to develop effective drug compounds and vaccines against strain-specific *B. suis*. Nevertheless, the study has some limitations as all methods were carried out on a computational approach. Furthermore, research into this study is required at the experimental level to confirm the findings. In vitro and in vivo studies are needed to make the analysis more robust.

## 5. Conclusions

Understanding the proteome of a pathogen is important as it facilitates further comprehensive analysis of proteins in various biochemical and pathological pathways that help in the identification of novel drug targets. Subtractive genomics can aid in arraying the vast information regarding genomics and proteomics of various pathogens, providing acceleration in drug designing and pharmacogenomics in the treatments of bacterial infection. Therefore, the current study applied the subtractive genomics approach for the prioritization of potent drug targets against the *B. suis* 1330 strain. It smears multiple essential analyses at different stages, i.e., non-homologs, essential, drug target-like, and unique to pathogens proteins. In this study, a number of proteins along with isocitrate lyase (ICL) was shortlisted as a novel drug target against *B. suis*. Consequently, the shortlisted essential proteins may be further studied and used as a therapeutic target by novel drug and vaccine candidates for *B. suis*. The present work uses a pharmacoinformatic approach to investigate the natural products ZINC library (n = 18,000) against ICL as a possible inhibitor. The six compounds, i.e., ZINC95543764, ZINC20115475, ZINC02688148, ZINC04232055, ZINC04231816, and ZINC04259566 were identified as potential inhibitors based on ligand–protein binding pattern (lowest estimated binding energy). However, the ADMET profiling showed that all compounds fulfill ADMET properties, except for ZINC20115475 showing positive Ames activity, whereas ZINC02688148, ZINC04259566, ZINC04232055, and ZINC04231816 showed hepatoxicity, while all compounds were observed to have no skin sensitization. In light of these parameters, we recommend the ZINC95543764 compound for further experimental studies. However, experimental validation with computational approaches is required for further analysis to improve the efficacy of predicted targets.

## Figures and Tables

**Figure 1 bioengineering-09-00633-f001:**
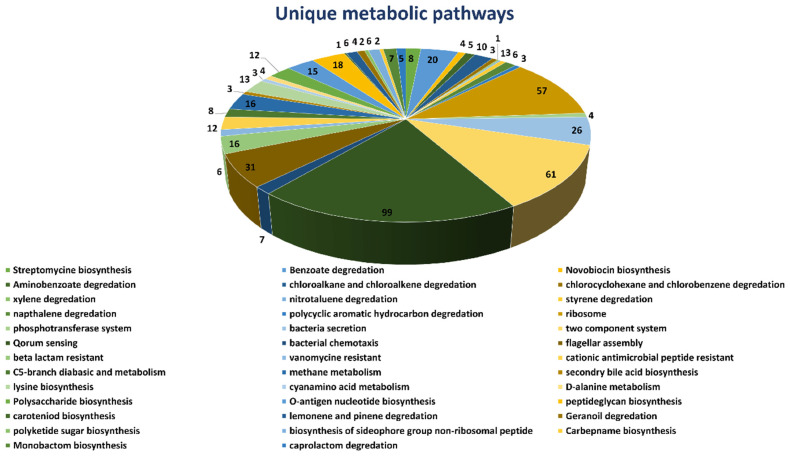
Unique metabolic pathways. Schematic representation of unique metabolic pathways found in *Brucella suis* along with the number of proteins identified in it.

**Figure 2 bioengineering-09-00633-f002:**
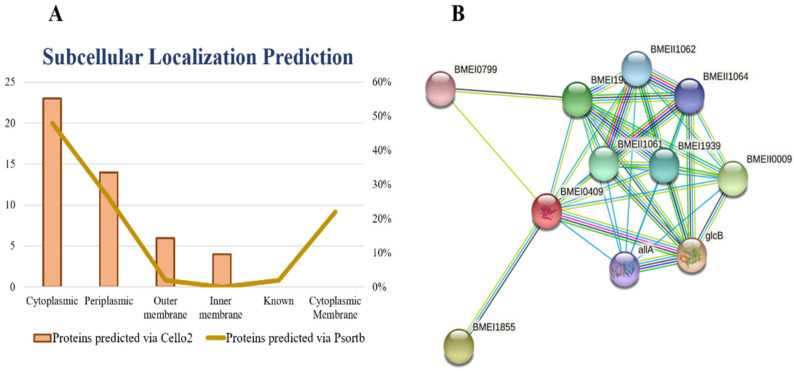
Subcellular localization and PPI. (**A**) Quantitative representation of subcellular localization of shortlisted essential, druggable, pathogen-specific proteins predicted through PSORTb, and Cello2 and (**B**) Protein–protein interaction analysis for ICL. The STRING results showed different interactions of proteins with each other represented by nodes and edges. The number of nodes represents proteins such as splice isoforms and post-translational modified proteins. Each node represents a specific protein encoded by a single gene. The color of nodes explains the protein interaction as first shell interactors and query protein are represented by colored nodes while the second shell interactors are represented by white nodes.

**Figure 3 bioengineering-09-00633-f003:**
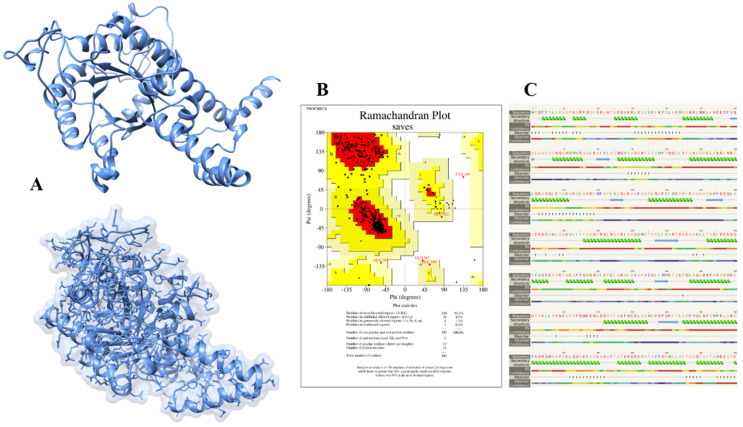
Modeled structure of isocitrate lyase. ICL modeled protein structure using a template-based approach. The predicted secondary structure of isocitrate lyase is exhibited by different signs and colors, such as: green helices represent alpha-helices, blue arrows indicate β-strands, a sign of interrogation (?) assigns a disordered region, and the paint line indicated the coil. The confidence of the predicted secondary structure is indicated by colors, red means high confidence, and blue color shows low confidence of secondary structure prediction (**A**) its validation via Ramachandran plot resulting in the structure validity for 90% and (**B**) secondary stricture prediction via Pyre2 tool (**C**).

**Figure 4 bioengineering-09-00633-f004:**
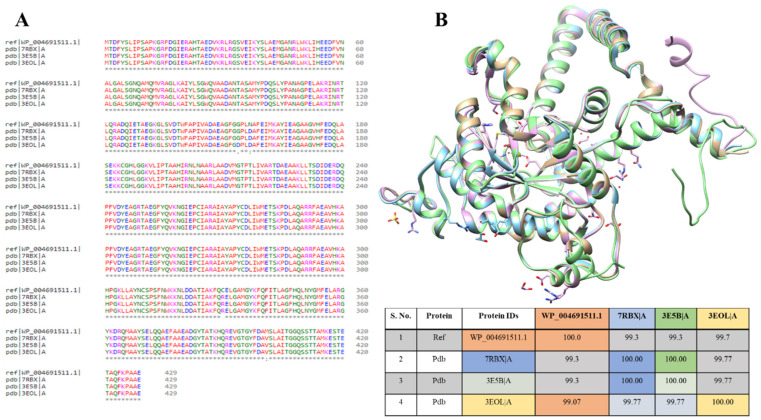
Multiple sequences alignment of ICL. (**A**) MSA of ICL from *B. suis* aligned with ICL protein from *B. abortus* 2308 (PDB ID: 7RBX), *B. melitensis* (PDB ID: 3E5B), and *B. melitensis* (P43212) (PDB ID: 3EOL) showing ~99.3% sequence similarity. (**B**) The superimposed structure of ICL proteins from B. suis (green color), *B. abortus* (blue), *B. melitensis* (tan), and *B. melitensis* P43212 (pink) showing the RMSD of ~2 Å among each other.

**Figure 5 bioengineering-09-00633-f005:**
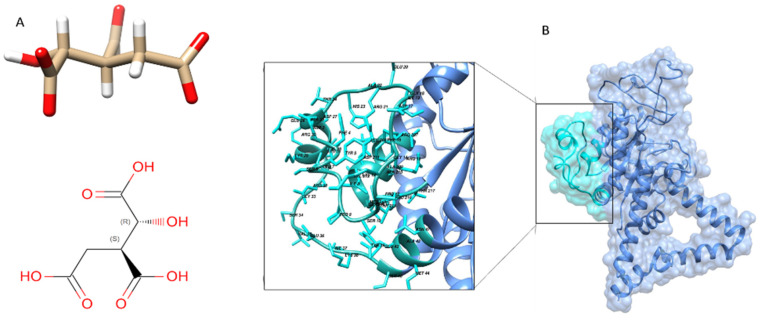
Ligand and active site prediction. Isocitric acid is predicted as a ligand via ProBis tool for isocitrate lyase (**A**), and the active site is predicted via MOE in isocitrate lyase protein (**B**).

**Figure 6 bioengineering-09-00633-f006:**
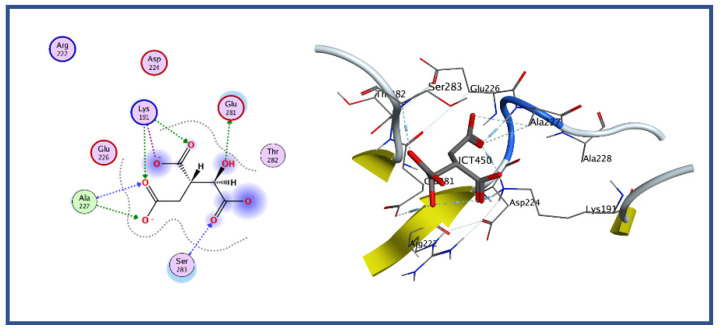
Docking analysis for ICT. Docking study for ICT against ICL active site showing binding affinity of −6.9 kcal/mol.

**Figure 7 bioengineering-09-00633-f007:**
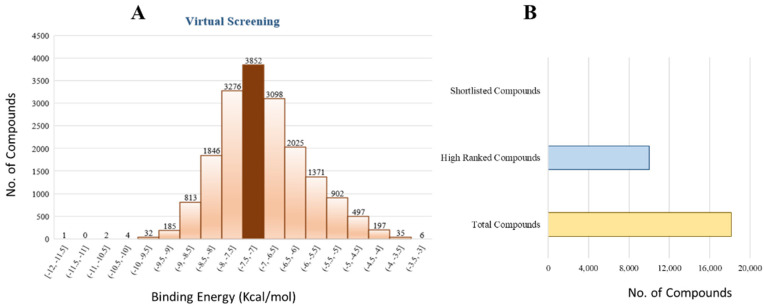
Virtual Screening: (**A**) Virtual screening studies for ICL using Zinc library of 18,000 compounds showing the binding energies ranging from the −3 to −12 kcal/mol. The resulted showed that about 3852 compounds docked at same position with the binding energies of −7 to −7.5 kcal/mol (brown color). (**B**) showing the overall summary of virtual screening from total of 18,00 compounds to 10, 000 high ranked compounds and finally resulted in the identification of six potential inhibitors i.e., ZINC95543764, ZINC02688148, ZINC20115475, ZINC04232055, ZINC04231816, and ZINC04259566.

**Figure 8 bioengineering-09-00633-f008:**
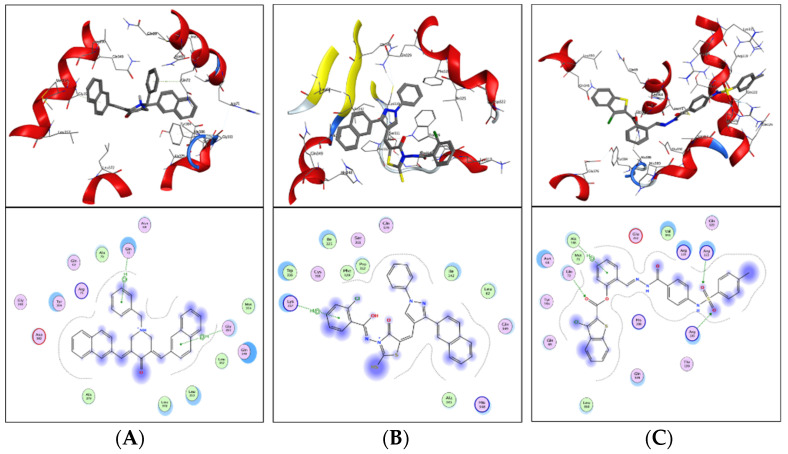
Molecular docking. Molecular docking analysis for the shortlisted compounds (**A**) ZINC20115475, (**B**) ZINC02688148, and (**C**) ZINC04231816.

**Figure 9 bioengineering-09-00633-f009:**
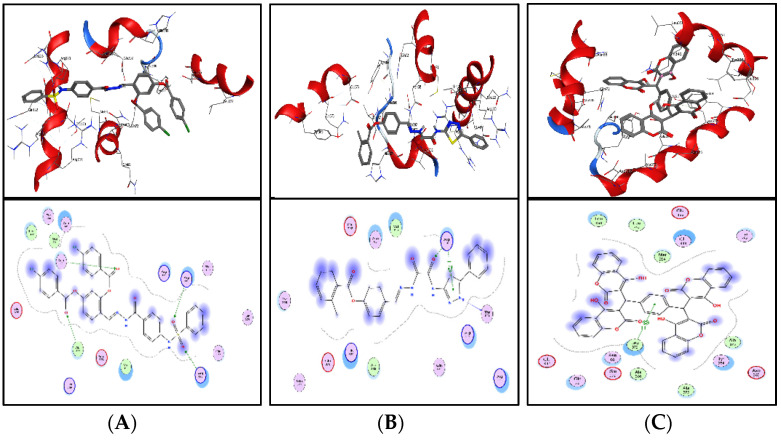
Molecular docking. Molecular docking analysis for the shortlisted compounds (**A**) ZINC04232055, (**B**) ZINC04259566, and (**C**) ZINC95543764.

**Figure 10 bioengineering-09-00633-f010:**
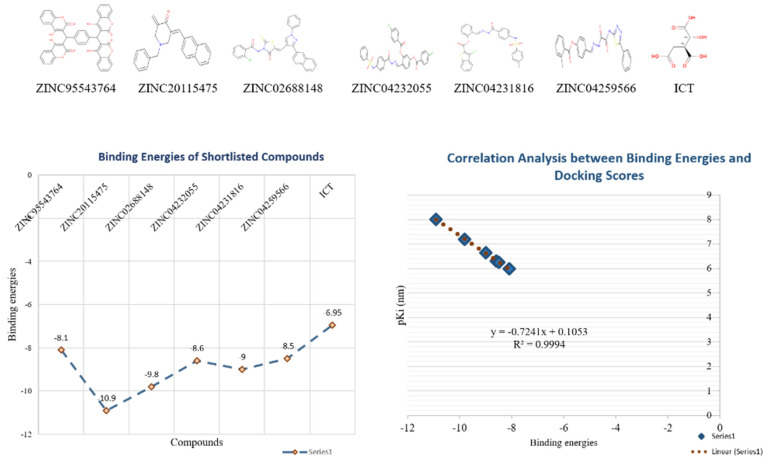
Correlation analysis for Ki and binding energies: the correlation analysis for six shortlisted compounds for ICL among Ki and binding energies. It shows the string regression correlation with R^2^ = 0.99.

**Figure 11 bioengineering-09-00633-f011:**
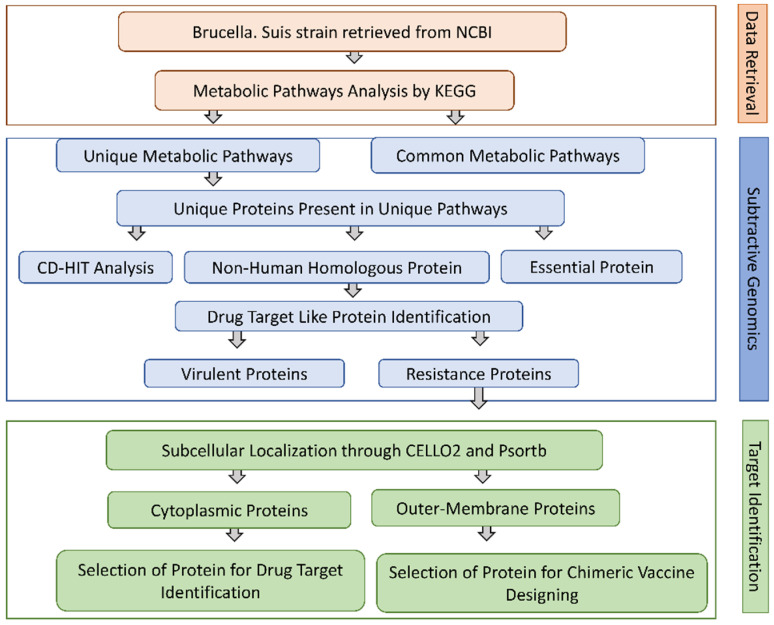
Flow chart for subtractive genomics. Flowchart of proposed study for drug target identification against *Brucella suis*.

**Table 1 bioengineering-09-00633-t001:** Subtractive genomic analysis scheme toward the identification of novel therapeutic targets against *B. suis*.

Sl. No.	Subtractive Approaches	Used Bioinformatics Server and Tools	Number of Proteins
1	The whole proteome of *B. suis* 1330	NCBI database	2974
2	Removed non-paralogous (>80% identical)	CD-Hit suite	0
3	Essential proteins involved only in unique metabolic pathways	KAAS at KEGG	503
4	Proteins nonhomologous to H. sapiens	BLASTp (E value 10^−3^)	421
5	Essential proteins	DEG 15.2 server (E value ≤ 10^−5^)	350
6	Novel drug target proteins non-homologous to ‘anti-targets’	using BLASTp (E value < 0.005, Identity < 25%, Query length > 30%)	114
7	Virulent proteins	Using BLASTp against VFDB (E value ≤ 10^−5^)	45
8	Resistant proteins	Using BLASTp against ARG-ANNOT V6 (E value ≤ 10^−5^)	42
9	Highly conserved protein	BLASTp	1 (Isocitrate lyase)

**Table 2 bioengineering-09-00633-t002:** Docking scores and identified bond types predicted through MOE tool for shortlisted compounds.

S. No.	Ligand	Receptor	Interaction	Distance (Å)	E (kcal/mol)
1	ZINC95543764				
	6-ring	CD1 LEU 372	pi-H	3.88	−0.8
2	ZINC20115475				
	6-ring	CG GLN 72	pi-H	3.63	−0.7
	6-ring	CA GLY 353	pi-H	3.59	−0.7
3	ZINC02688148				
	6-ring	CB LYS 317	pi-H	3.78	−0.7
4	ZINC04232055				
	O 21	CB ARG 119	H-acceptor	3.62	−0.5
	O 22	NH2 ARG 123	H-acceptor	3.13	−3.7
	O 31	CB ALA 386	H-acceptor	3.37	−0.5
	O 41	NE2 GLN 72	H-acceptor	3.03	−2.9
5	ZINC04231816				
	O 13	NH1 ARG 116	H-acceptor	3.14	−1.3
	O 14	NE ARG 123	H-acceptor	3.15	−2.0
	O 14	NH2 ARG 123	H-acceptor	2.98	−4.0
	O 32	NE2 GLN 72	H-acceptor	2.91	−3.9
	6-ring	CB ALA 386	pi-H	3.91	−0.5
6	ZINC04259566				
	O 25	NH2 ARG 116	H-acceptor	3.13	−0.8
	N 28	CA THR 120	H-acceptor	3.26	−0.5
	5-ring	NH1 ARG 116	pi-cation	3.25	−1.1

**Table 3 bioengineering-09-00633-t003:** The predicted binding affinities and Ki values for compounds.

Compounds	Binding Affinity (kcal/mol)	Predicted Ki	pKi
ZINC95543764	−8.1	1.02 µm	5.9914
ZINC20115475	−10.9	9.87 nm	8.005683
ZINC02688148	−9.8	64.48 nm	7.190575
ZINC04232055	−8.6	493.49 nm	6.306722
ZINC04231816	−9.0	229.47 nm	6.639274
ZINC04259566	−8.5	557.52 nm	6.25374

**Table 4 bioengineering-09-00633-t004:** ADMET profiling for shortlisted compounds.

Name	Water Solubility (log mol/L)	CaCo2 Permeability (log Papp in 10^−6^ cm/s)	HIA (% Absorbed)	Skin Permeability (log Kp)	BBB Permeability (log BB)	Structure
ZINC95543764	−2.862	−0.025	90.747	−2.735	No	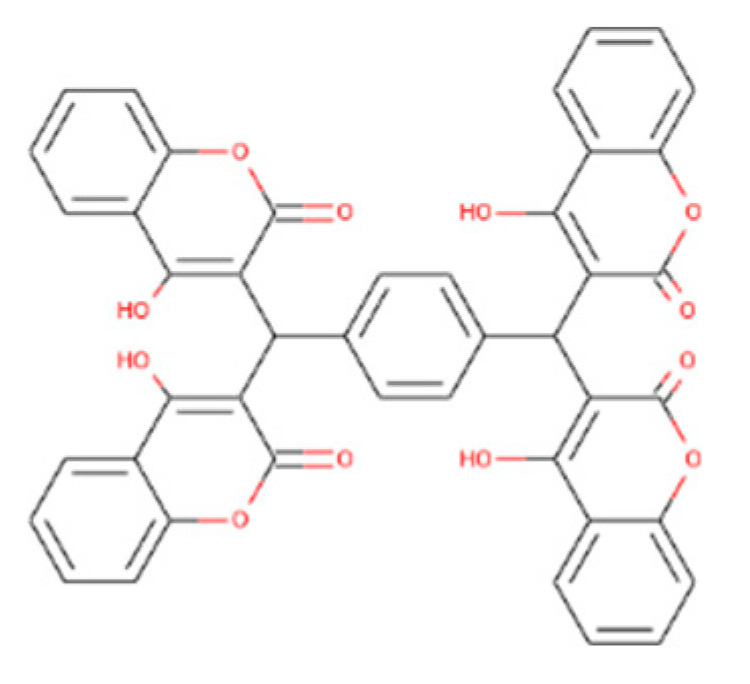
ZINC20115475	−4.486	0.875	92.679	−2.831	No	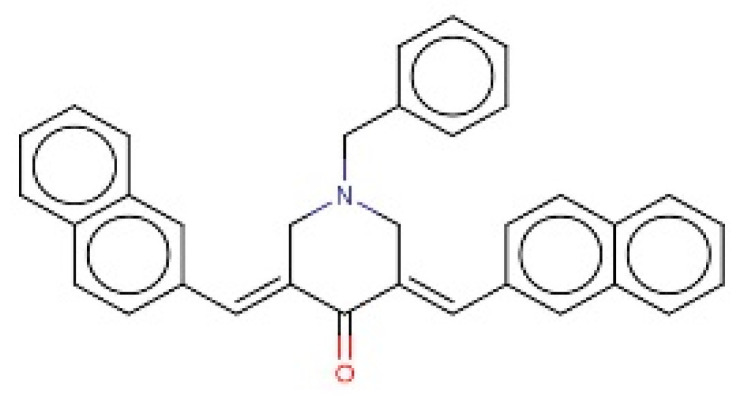
ZINC02688148	−5.503	0.279	92.894	−2.735	No	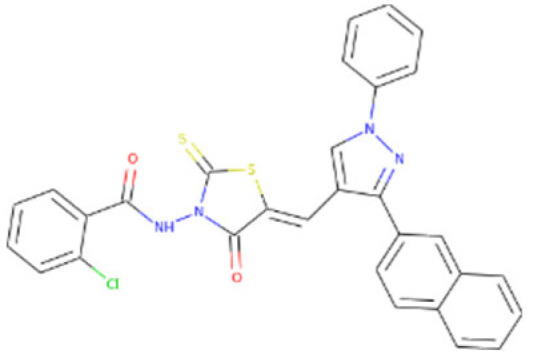
ZINC04232055	−3.067	0.182	89.573	−2.735	No	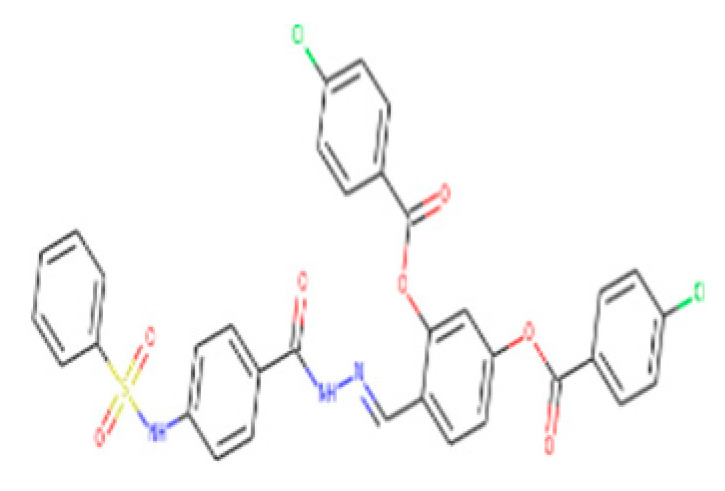
ZINC04231816	−4.069	0.864	100	−2.735	No	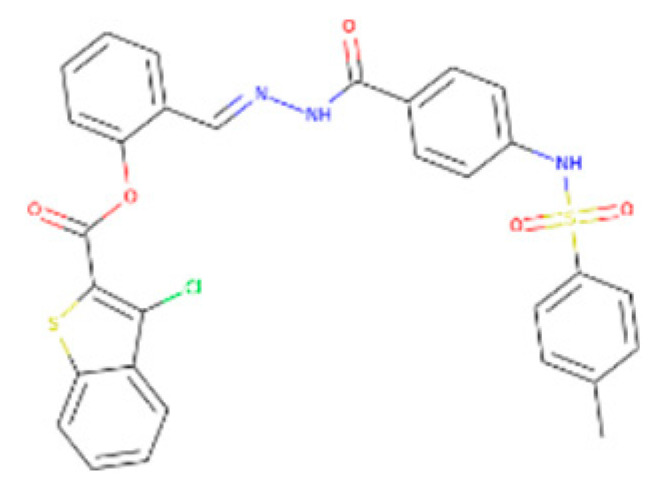
ZINC04259566	−4.241	0.94	75.892	−2.735	No	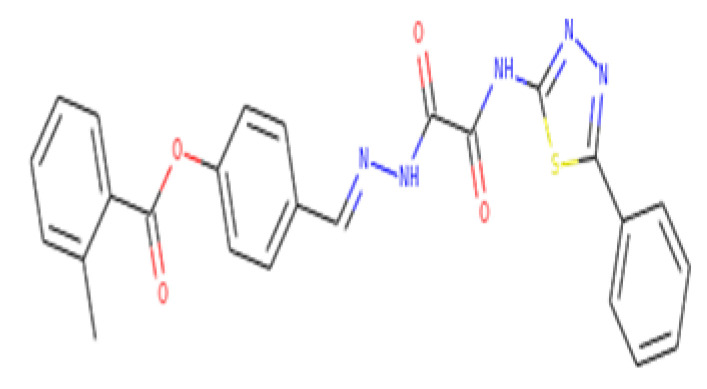

**Table 5 bioengineering-09-00633-t005:** Toxicity analysis of shortlisted compound against ICL.

Name	Max. Tolerated Dose (Human) (log mg/kg/day)	Minnow Toxicity (log mM)	*T. Pyriformis* Toxicity (log ug/L)	Oral Rat Acute Toxicity (LD_50_) (mol/kg)	Ames Test	Hepatotoxic	Skin Sensitization
ZINC95543764	0.438	−0.815	0.285	2.473	No	No	No
ZINC20115475	0.481	−1.927	0.367	2.312	Yes	No	No
ZINC02688148	0.571	−2.841	0.285	3.173	No	Yes	No
ZINC04232055	0.352	−1.6	0.285	2.828	No	Yes	No
ZINC04231816	0.411	−1.781	0.285	2.981	No	Yes	No
ZINC04259566	0.807	0.319	0.285	2.535	No	Yes	No

**Table 6 bioengineering-09-00633-t006:** Conservancy analysis isocitrate lyase (ICL); (BLASTp results against other *B. suis* strains).

S. No.	Sequence ID	Organism	Identity (%)
1	WP_004691058.1	*Brucella*	99.77
2	WP_006071281.1	*Brucella suis*	99.53
3	WP_006201131.1	*Brucella suis*	99.30
4	WP_004689882.1	*Brucella*	99.53
5	WP_006279211.1	*Brucella suis*	99.30
6	WP_004684105.1	*Brucella*	99.07

**Table 7 bioengineering-09-00633-t007:** Complete Proteome of Human-Host and *Brucella suis*.

Strain ID	Strain Name	Proteins
GCF_0000007505.1	1330	2974
9606	Human	~200,000
*Brucella suis*	bms	109 pathways
Human host	hsa	330 pathways

## Data Availability

Not applicable.

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
