# Peer review of "Integrated Bioinformatics-Based Subtractive Genomics Approach to Decipher the Therapeutic Drug Target and Its Possible Intervention against Brucellosis"

_bioengineering, 2022, doi:10.3390/bioengineering9110633_

Round 1

Reviewer 1 Report

This study, which seeks to detect drugs against brucellosis using computational biology, seems interesting because, starting from the proteome of the pathogenic bacteria, it achieves its goal in 16 steps.

This article therefore provides a lot of very educational information (too much?) and shows the different facets of drug research using computational biology. On the other hand, some of the information is perhaps a little too academic (Example: line 90-100 or line 187-190....).

I think that the content will have to be reinvested, once the form has been reworked. Indeed, in my opinion, a lot needs to be done: explain the acronyms, revise the English, keep the same acronyms throughout the text, make complete sentences, redistribute the paragraphs (i.e.: typical information in the captions remains in the text of the results). Give a clear strategy and propose a supported discussion and perspectives. You will find below some very specific comments that could help to restructure the article.

Abstract

Line 40: At that point, we don’t understand clearly how they found “Isocitrate Lyase”.

Introduction

Line 79: the virulence explanation by this factor is not enough clear

Line 85: The subject here is clearly explained

Line 90-100: a bit too long and not really important, we now know that bioinfo and computational biology are necessary

Results and discussion

This section would be more readable if the paragraph numbering would correspond to the line numbering of table 2

Table 2 and Figure 2: I don’t understand why there are no table1 and Figure 1: did I lost something? 

Line 226-229: The target that has been highlighted in this work has already been studied as a potential target against different pathogens, which impoverishes this study. Indeed, in general, to find new drugs against a pathogen, existing drugs against other pathogens are tested in the laboratory. 

Line 245: reformulate this sentence « Number of nodes characterized proteins such as post-translational modified proteins, splice isoform whereas each node defined specific protein encoded by a single gene »

Line 243-249: this text should rather be the subject of a figure legend or materials and methods

Paragraph 2.11: In the PDB database, there are several three-dimensional structures of isocitrate lyase. It would be interesting to look at them and perhaps try a multiple alignment on these already existing 3D structures. In particular, there is the 7RBX form brucella Melitensis: would this not be a suitable structure to use? The systematic study of all the structures of the isocitrate lyase of the PDB is to be considered.

Line 290-296: must go in legend or M&M

Figure 5: I don’t understand this sentence “potential inhibitor predicted for ICL as Isocitric acid through probis (A), »

Figure 5B: the magnified part seems not to be in the same plane as the non magnified part. Also I don't see the docked ligand in the active site in this figure. It should be put in a different colour. 

Line 351: To understand the importance of this score, the authors should give us additional information such as min/max of this score for example.

Figure 6: residue 224? it is not clear which residue it is. The results in figure 6 should be better explained: what bonds were found? H-bonding, salt bridges?

Line 359: write a little introduction: why should we use a zinc library?

Figure 7: you should explain the coordinate of the graph

Paragraph 2.14

How did you get from the list of 10000 molecules to the short list? Can we trust scores that evaluate so many molecules at once? The different binding between protein and possible inhibitor are well explained.

I think, shortlisted selected compounds are competitive inhibitors (bound the same binding site than the physiological ligand)

Figure 8 and Figure 9. Could be a good idea to put them on the same figure.

Line 446: could you define BBB analyses?

Line 449: Is Da more appropriate than Amu to define atomic mass or even g/mol?

Line 451: could you define what is RO5?

Line 471 and line 482: Ames or AMES?

Line 493: I don’t understand this sentence

The discussion part seems to be reduced to a trickle.

Author Response

Reviewer 1 Comments and Suggestions for Authors

This study, which seeks to detect drugs against brucellosis using computational biology, seems interesting because, starting from the proteome of the pathogenic bacteria, it achieves its goal in 16 steps.

This article therefore provides a lot of very educational information (too much?) and shows the different facets of drug research using computational biology. On the other hand, some of the information is perhaps a little too academic (Example: line 90-100 or line 187-190....). I think that the content will have to be reinvested, once the form has been reworked. Indeed, in my opinion, a lot needs to be done: explain the acronyms, revise the English, keep the same acronyms throughout the text, make complete sentences, redistribute the paragraphs (i.e.: typical information in the captions remains in the text of the results). Give a clear strategy and propose a supported discussion and perspectives. You will find below some very specific comments that could help to restructure the article.

Authors response: We are thankful to the reviewer for evaluating our manuscript thoroughly. The manuscript is now revised and updated as per suggestion. The recommended changes are made in red font in revised manuscript.

Abstract

Line 40: At that point, we don’t understand clearly how they found “Isocitrate Lyase”.

Authors Response: Thanks to the reviewer for pointing out this, actually the Drug Target Iso-citrate Lyase was finalized after the extensive literature survey as previously reported as a promiscuous drug target against tuberculosis. Isocitrate Lyase as is the first enzyme involved in glyoxylate cycle. Many plants and microorganisms are relying on glyoxylate cycle enzymes to survive upon downregulation of tricarboxylic acid cycle (TCA cycle). The point was added in the abstract as per suggestion.

Introduction

Line 79: the virulence explanation by this factor is not enough clear

Authors Response: The sentence is now corrected 

Line 85: The subject here is clearly explained

Authors Response: It is now corrected

Line 90-100: a bit too long and not really important, we now know that bioinfo and computational biology are necessary

 Authors Response: The paragraph is now shortened as per reviewer suggestion. 

Results and discussion

This section would be more readable if the paragraph numbering would correspond to the line numbering of table 2

Authors Response: It is now corrected

Table 2 and Figure 2: I don’t understand why there are no table1 and Figure 1: did I lost something? 

Authors Response: The Table1 and Figure 1 are present in the methodology section on page no 18.

Line 226-229: The target that has been highlighted in this work has already been studied as a potential target against different pathogens, which impoverishes this study. Indeed, in general, to find new drugs against a pathogen, existing drugs against other pathogens are tested in the laboratory. 

Authors Response: Thanks for highlighting this point. We though agree to the reviewer that the identified drug target is not innovative, however the idea is unique to find potential drug targets that is previously not identified against the Brucella suis which is largely ignored topic of interest within the scientific community. We paid particular attention to the unexplored genomics data which was earlier abandoned. Moreover, the usage of isocitrate lyase as potential drug target against other pathogen support the identified computational studies for Brucella and thus strengthen our study.

Line 245: reformulate this sentence « Number of nodes characterized proteins such as post-translational modified proteins, splice isoform whereas each node defined specific protein encoded by a single gene »

Authors Response: The mentioned sentence has been corrected as per reviewer suggestion in the revised manuscript and move to the figure legend in Figure 3B.

Line 243-249: this text should rather be the subject of a figure legend or materials and methods

Authors Response: It is now moved to Figure 3B legends

Paragraph 2.11: In the PDB database, there are several three-dimensional structures of isocitrate lyase. It would be interesting to look at them and perhaps try a multiple alignment on these already existing 3D structures. In particular, there is the 7RBX form Brucella Melitensis: would this not be a suitable structure to use? The systematic study of all the structures of the isocitrate lyase of the PDB is to be considered.

Authors Response: We are thankful to the reviewer for highlighting this. As per the suggestion we have added comparative analysis of these structure and provided under “2.11. Comparative Structure Homology Modeling and Validation” and Figure 5.

Line 290-296: must go in legend or M&M

Authors Response: Moved to the legend as per suggestion

Figure 5: I don’t understand this sentence “potential inhibitor predicted for ICL as Isocitric acid through probis (A), »

Authors Response: The sentenced is now corrected in the revised manuscript

Figure 5B: the magnified part seems not to be in the same plane as the non-magnified part. Also I don't see the docked ligand in the active site in this figure. It should be put in a different colour. 

Authors Response: The figure is now corrected in the revised manuscript. Figure 5 (Now fig 6) only showed the active site of protein and its active residues and predicted ligand. However, the ligand in the biding site of protein is shown in figure 6 (Now fig 7)

Line 351: To understand the importance of this score, the authors should give us additional information such as min/max of this score for example.

Authors Response: We are thankful to the reviewer, this sentenced is now modified in the revised manuscript as per the reviewer suggestion.

Figure 6: residue 224? it is not clear which residue it is. The results in figure 6 should be better explained: what bonds were found? H-bonding, salt bridges?

Authors Response: The Figure 6 is now improved, the residue interaction can be seen now. The interaction detail of the ligand and protein is also added in the revised manuscript.

Line 359: write a little introduction: why should we use a zinc library?

Authors Response: The introduction is updated as per suggestion (Line 101-106)

Figure 7: you should explain the coordinate of the graph

Authors Response: Revised the figure as per suggestion

Paragraph 2.14

How did you get from the list of 10000 molecules to the shortlist? Can we trust scores that evaluate so many molecules at once? The different binding between protein and possible inhibitors are well explained.

Authors Response: The 10,000 compounds were selected based on their lower binding energies compared to the reference compound (already explain in “2.13. Molecular Docking Study Virtual Screening” paragraph. We agree to the reviewer that just binding energy are not enough to examine any compound, however, it is a good practice to narrow down the potent ones from the bulk of compounds such as 18,000 in our case to only examine the high-affinity compound for further wet lab experimentation.

I think, shortlisted selected compounds are competitive inhibitors (bound the same binding site as the physiological ligand)

Authors Response: Actually, the same binding pocket was selected to dock the library to evaluate their possible binding interaction with protein active amino acids (as the active binding site of protein is already defined). However, it is not linked to the competitive or non-competitive nature of ligand.

Figure 8 and Figure 9. Could be a good idea to put them on the same figure.

Authors Response: We respect your suggestion, if we merge these figures, the resolution might compromise.

Line 446: could you define BBB analyses?

Authors Response: Done as per suggestion

Line 449: Is Da more appropriate than Amu to define atomic mass or even g/mol?

Authors Response: In most cases, the number of nucleons A in an atom's nucleus is close to, but does not exactly equal, the atomic mass in daltons. Molar mass (in grammes per mole) is proportional to the number of nucleons on average in a given molecule. A carbon-12 atom has 12 daltons of mass since that is the number of nucleons it has (6 protons and 6 neutrons). However, the binding energy of the nucleons in an object's atomic nucleus, as well as the mass and binding energy of its electrons, all play a role in determining the object's overall mass on the atomic scale. Therefore, this equality holds only for the carbon-12 atom in the stated conditions and will vary for other substances. That is why Da is more appropriate than Amu to define atomic mass or even g/mol for large-sized molecules.  However, It is now corrected in the revised mansucript

Line 451: could you define what is RO5?

Authors Response: Explained as per suggestion

Line 471 and line 482: Ames or AMES?

Authors Response:  Thanks to the reviewer for pointing out these mistakes, we now replaced AMES with Ames.

Line 493: I don’t understand this sentence

Authors Response: it is now revised

The discussion part seems to be reduced to a trickle.

Authors Response: It is now added to the revised manuscript.

Reviewer 2 Report

The authors have presented a detailed evaluation and very extensive work and are to be congratulated for their study.

I have few points to comment upon, as, in general, the work has been performed satisfactorily.

However, the manuscript needs extensive restructuring before acceptance.

As a start, the manuscript is very long and tiresome to authors.

I understand the extent of the work performed and the authors should receive credit for that, but really such a long manuscript is not helpful at all. The authors must either reduce the amount of data in there or alternatively they can consider a second paper (preferable).

Moreover, I do not like the approach of merging the results with discussion. I see the usefulness, but in this approach the comments about the findings are limited really and rather shallow comparatively to the extent of the work. The two sections must be separated, and the discussion must be extended. Significant relevant references are missing. A new sub-section must be introduced to indicate the clinical consequences of the study.

The above must be performed and a revised manuscript must be submitted for new detailed evaluation before possible acceptance.

Author Response

Reviewer 2 Comments and Suggestions for Authors

The authors have presented a detailed evaluation and very extensive work and are to be congratulated for their study.

I have few points to comment upon, as, in general, the work has been performed satisfactorily.

However, the manuscript needs extensive restructuring before acceptance.

As a start, the manuscript is very long and tiresome to authors.

I understand the extent of the work performed and the authors should receive credit for that, but really such a long manuscript is not helpful at all. The authors must either reduce the amount of data in there or alternatively they can consider a second paper (preferable).

Authors Response: Thank you for the suggestion. However, the work is not enough for two publications as it totally based on in-silico study.

Moreover, I do not like the approach of merging the results with discussion. I see the usefulness, but in this approach the comments about the findings are limited really and rather shallow comparatively to the extent of the work. The two sections must be separated, and the discussion must be extended. Significant relevant references are missing. A new sub-section must be introduced to indicate the clinical consequences of the study.

Authors Response: We appreciate the reviewer for working on the manuscript. The discussion part is now provided in the revise manuscript.

 The above must be performed and a revised manuscript must be submitted for new detailed evaluation before possible acceptance.

Authors Response: The suggested revision is performed and the manuscript is now updated based on the suggested comments.

Round 2

Reviewer 1 Report

Dear authors,

Thank you for having made the effort to smooth out your manuscript to allow future readers, who I hope will be many, to have a better readability.

Author Response

Dear authors,

Thank you for having made the effort to smooth out your manuscript to allow future readers, who I hope will be many, to have a better readability.

Author Response: We appreciate the reviewer for working on our manuscript.

Reviewer 2 Report

The revised manuscript has been improved.

The authors need to add an extra paragraph in the discussion about the clinical implications of the work and then it can be published.

Author Response

Reviewer 2: Comments and Suggestions for Authors

The revised manuscript has been improved.

The authors need to add an extra paragraph in the discussion about the clinical implications of the work and then it can be published.

Author Response: We are thankful to the reviewer for reviewing the manuscript. The suggested paragraph is added in the revised manuscript in blue text font.
